# Recurrence Patterns for Pancreatic Ductal Adenocarcinoma after Upfront Resection Versus Resection Following Neoadjuvant Therapy: A Comprehensive Meta-Analysis

**DOI:** 10.3390/jcm9072132

**Published:** 2020-07-06

**Authors:** Bathiya Ratnayake, Alina Y. Savastyuk, Manu Nayar, Colin H. Wilson, John A. Windsor, Keith Roberts, Jeremy J. French, Sanjay Pandanaboyana

**Affiliations:** 1Department of Surgery, Faculty of Medical and Health Sciences, University of Auckland, Auckland 1010, New Zealand; Crat791@aucklanduni.ac.nz (B.R.); alina.savastyuk@gmail.com (A.Y.S.); j.windsor@auckland.ac.nz (J.A.W.); 2Department of Gastroenterology, Freeman Hospital, Newcastle upon Tyne NE7 7DN, Tyne and Wear, UK; manu.nayar@nhs.net; 3Department of Hepatobiliary, Pancreatic and Transplant Surgery, Freeman Hospital, Newcastle upon Tyne NE7 7DN, Tyne and Wear, UK; Colin.Wilson@nhs.net (C.H.W.); Jeremy.French@nhs.net (J.J.F.); 4Department of Immunology and Immunotherapy, University of Birmingham, Birmingham B15 2TT, UK; Keith.Roberts@uhb.nhs.uk

**Keywords:** recurrence, neoadjuvant chemotherapy, pancreatic ductal adenocarcinoma, pancreatic surgery

## Abstract

Background: Neoadjuvant therapy (NAT) represents a paradigm shift in the management of patients with pancreatic ductal adenocarcinoma (PDAC) with perceived benefits including a higher R0 rate. However, it is unclear whether NAT affects the sites and patterns of recurrence after surgery. This review seeks to compare sites and patterns of recurrence after resection between patients undergoing upfront surgery (US) or after NAT. Methods: The EMBASE, SCOPUS, PubMed, and Cochrane library databases were systematically searched to identify eligible studies that compare recurrence patterns between patients who had NAT (followed by resection) with those that had US. The primary outcome included site-specific recurrence. Results: 26 articles were identified including 4986 patients who underwent resection. Borderline resectable pancreatic cancer (BRPC, 47% 1074/2264) was the most common, followed by resectable pancreatic cancer (RPC 42%, 949/2264). The weighted overall recurrence rates were lower among the NAT group, 63.4% vs. 74% (US) (OR 0.67 (CI 0.52–0.87), *p* = 0.006). The overall weighted locoregional recurrence rate was lower amongst patients who received NAT when compared to US (12% vs. 27% OR 0.39 (CI 0.22–0.70), *p* = 0.004). In BRPC, locoregional recurrence rates improved with NAT (NAT 25.8% US 37.7% OR 0.62 (CI 0.44–0.87), *p* = 0.007). NAT was associated with a lower weighted liver recurrence rate (NAT 19.4% US 30.1% OR 0.55 (CI 0.34–0.89), *p* = 0.023). Lung and peritoneal recurrence rates did not differ between NAT and US cohorts (*p* = 0.705 and *p* = 0.549 respectively). NAT was associated with a significantly longer weighted mean time to first recurrence 18.8 months compared to US (15.7 months) (OR 0.18 (CI 0.05–0.32), *p* = 0.015). Conclusion: NAT was associated with lower overall recurrence rate and improved locoregional disease control particularly for those with BRPC. Although the burden of liver metastases was less, there was no overall effect upon distant metastatic disease.

## 1. Introduction

Patients who have undergone potentially curative treatment for pancreatic ductal adenocarcinoma (PDAC) typically suffer treatment failure with cancer recurrence [1], frequently within two years of resection [2]. Recurrence, including multi-site recurrence and liver metastases, are key prognosticators of overall survival [3]. Several risk factors for specific recurrence patterns have been identified. Positive resection margin and poor tumour differentiation [2,4] are risk factors for locoregional recurrence. Tumour size, elevated preoperative Ca19-9 serum level, and perineural invasion are risk factors for distant recurrence [5].

While upfront surgery followed by adjuvant therapy has been the standard of care for PDAC, this is being challenged by recent evidence with use of neoadjuvant chemotherapy (NAT) for borderline resectable pancreatic cancer (BRPC) [6] and resectable (RPC) [7] pancreatic cancer (LAPC) [8]. It is important to determine the impact of different treatment strategies on pattern of disease recurrence [9,10]. NAT is associated with a higher rate of achieving R0 margins [11,12] and may control micrometastatic disease; thus, it is important to determine whether NAT influences patterns of treatment failure in terms of the frequency and site of recurrent cancer.

While here have been meta-analyses examining the impact of NAT on overall survival [13,14], there is paucity of evidence examining the impact of NAT and upfront surgery (US) on disease recurrence and recurrence patterns with and without neoadjuvant chemotherapy.

This present study aimed to undertake a meta-analysis of available evidence to compare patients with PDAC undergoing upfront surgery versus neoadjuvant therapy followed by surgery to review sites of recurrence, recurrence-free survival patterns, overall recurrence rate, and time to recurrence in patients with resectable, borderline resectable and locally advanced pancreatic cancer.

## 2. Materials and Methods

The protocol of this review was registered on the Prospero database prior to inception (CRD42019159031).

### 2.1. Literature Search

The EMBASE, SCOPUS, PubMed and Cochrane library databases were systematically searched with the aid of a pre-devised literature search strategy that followed the Preferred Reporting Items for Systematic Reviews and Meta-Analyses (PRISMA) guidelines [15]. Details of the query string with accompanying Boolean operators are reported in Appendix B. Limits were set to English studies reported from January 2000 to February 2020 in human patients. Reference lists of included studies were then examined by two authors (C.B.B.R. and A.Y.S.) to identify eligible studies missed by the original search strategy.

### 2.2. Data Selection, Extraction, Risk of Bias Assessment and Publication Bias Assessment

Studies were included in the quantitative meta-analysis if they reported comparative recurrence outcomes specific to patients who underwent NAT compared to those undergoing US for PDAC. More specifically, if they reported overall recurrence, time to first recurrence, sites of recurrence and/or recurrence free survival (RFS). Studies were excluded if they were case reports/series, conference abstracts, study protocols, non-comparative observational cohorts, did not report recurrence or RFS data for either NAT or upfront surgery patients, did not employ NAT, and articles published prior to 2000. Title, abstract and full text screening were performed by two investigators (C.B.B.R. and A.Y.S.). Consistency of data extraction was compared, and disputes were adjudicated by a third author (S.P.). Baseline characteristic data including: age; gender; location of tumour; preoperative tumour stage and resectability; details of NAT data; preoperative radiotherapy; adjuvant therapy; pathological outcome data including; nodal status; tumour differentiation; tumour size; perineural/lymphovascular invasion; R0 resection; and patterns of recurrence including overall recurrence rate, time to first recurrence, site of recurrence, and RFS. The primary outcome was site-specific recurrence. The secondary outcomes were overall recurrence, time to first recurrence, and recurrence-free survival.

### 2.3. Assessment of Study Quality

The risk of bias and study quality assessment was performed through use of the methodological index for non-randomized studies (MINORS) grading criteria [16] for non-randomised studies. The Cochrane’s risk of bias tool [17] was used for randomised controlled trials. Publication bias was identified through visual inspection of funnel plots whereby bias existed if individual studies lay beyond the funnel limits.

### 2.4. Terminology and Definitions

*NAT* included preoperative chemotherapy with or without radiotherapy. Sole radiotherapy-based NAT regimens were not assessed in this meta-analysis. Fluoropyrimidine-based chemotherapies included S-1 (5-fluorouracil prodrug tegafur with oteracil, and gimeracil) [18], capecitabine and 5-fluorouracil; platinum-based chemotherapies included cisplatin and oxaliplatin; and plant alkaloids included docetaxel and paclitaxel. FOLFIRINOX was a combination regime of 5-fluorouracil, leucovorin, oxaliplatin, and irinotecan. If an agent was used in combination with another, this was considered combination chemotherapy. Upfront surgery (US) included all forms of pancreatic resection (including pancreatoduodenectomy, distal pancreatectomy and total pancreatectomy) performed shortly after the diagnosis of a pancreatic cancer without preoperative NAT. The authors identified resectable (RPC), borderline resectable (BRPC), and locally advanced (LAPC) when reported as such. The definitions of these categories were variably reported and have changed over the course of the study with the publication of multiple definitions [19,20,21]. Nodal status was considered N0 when there were no positive nodes at the time of surgery. Tumour differentiation was based on histology of the specimen following resection [22] and tumour size by the largest diameter reported by the pathologist. Presence of malignant cells in neural tissue or lymphovascular tissue denoted perineural or lymphovascular invasion [23]. R0 resection was defined when there was no microscopic tumour present within 1 mm of the resection margin [24]. Recurrence patterns included overall recurrence rates, time to first recurrence and recurrence-free survival rates. Overall recurrence was defined by the first presentation of previously unknown disease activity during postoperative follow-up [25]. Locoregional recurrence was the presence of tumour recurrence at the site of previous resection or in surrounding lymph nodes (LN). Distant recurrence referred to recurrence present in distant organs. RFS rate was defined as the proportion of patients who had no confirmed recurrence of cancer and included the terms progression-free survival and disease-free survival [26]. Time to first recurrence represented the interval following resection that the first recurrence of disease was observed and was calculated by the mean RFS [27]. Disease-specific survival [28,29] or cancer-specific survival [8] were not included in this analysis as these terms focused solely on survival secondary to the cancer.

### 2.5. Statistical Analysis

The meta-analysis was conducted in its entirety with the packages: tidyverse [30], meta [31], metaphor [32], and MetaAnalyser (Jack Bowden and Christopher Jackson, UK) [33] in R project (R Foundation for Statistical Computing, Austria 2014). A Mantel-Haenszel model was utilised to perform the pairwise meta-analysis. The output was reported as odds ratios (OR) or standard mean difference (SMD) with 95% confidence intervals (CI). I^2^ test allowed the authors to determine statistical heterogeneity whereby a threshold of 50% suggested moderate heterogeneity and 75% indicative of substantial heterogeneity [34]. Weighted means and recurrence rate estimates with their corresponding 95% CI were calculated for age, tumour diameter, follow-up interval, and recurrence outcomes. Weighted means were calculated by the inverse variance method. Weighted recurrence rate estimates were determined by a random intercept logistic regression model [31]. Weighted estimates were only calculated when three or more datapoints were available. WebPlotDigitizer (Ankit Rohatgi, CA, USA, 2019) [35] was employed to extract raw data from reported Kaplan Meier survival curves if articles failed to provide raw numbers for RFS or time to first recurrence. The median (1st and 3rd quartile) time to first recurrence was only calculable from a Kaplein Meier curve if both survival curves crossed the 75% and 25% survival threshold. Mean estimates were then derived from median and ranges or quartiles through Wan et al. [36].

Propensity score matched data was used in preference to raw unmatched data to improve homogeneity of the datasets in relevant studies [8,27,37,38]. However, if baseline characteristic data or outcome data were not reported for propensity score matching, the unmatched data was used [28]. Historic [25] or unpublished [39] US cohorts were utilised if baseline characteristics were reported. Data for baseline characteristics was commonly reported for the entire study population irrespective of those who subsequently underwent resection. However, the remaining outcomes including pathological, resectability and recurrence data are reported for the population who ultimately underwent surgery and were subsequently followed-up. When calculating NAT agent-specific time to first recurrence, the authors included situations where the agent was used as the base NAT, used in combination with another agent or used in more than 30% of the cohort [40]. Recurrence outcomes were reported with significantly fewer patients than in the pathological and resectability outcomes in another study [41] and this is reflected in the meta-analysis.

## 3. Results

From 3933 potentially relevant studies, 22 were included in the quantitative analysis. Four additional studies [8,40,42,43] were recruited into the meta-analysis following a review of the reference lists of the 22 studies (Figure 1).

Among 4986 patients, 1699 received NAT followed by pancreatic resection (34%, 1699/4986) and 3287 patients US (66%, 3287/4986). The studies were from predominantly single centers (81%, 21/26) in Asia (42%, 11/26) and retrospective in design (81%, 21/26) (characteristics of all studies are summarised in Table 1).

Baseline characteristic data was reported for all patients initially recruited into the studies before NAT was administered (*n* = 5328). Patients were commonly female (54%, 2885/5328) in their seventh decade of life (weighted mean age 62 years, CI 60.4–63.6). Tumour location was reported in 20 studies [6,7,8,26,29,37,38,39,40,41,42,44,45,47,48,49,50,51,53,54] and was predominantly the pancreatic head or neck (79%, 2478/3147) (Appendix A).

### 3.1. Neoadjuvant and Adjuvant Therapy

Most patients received Gemcitabine-based NAT (24 studies [6,7,8,25,26,27,28,29,39,40,41,42,43,44,45,46,47,48,49,50,51,52,53,54], 62%, 851/1382). Fluoropyrimidines (20%, 282/1382) and FOLFIRINOX (13%, 182/1382) based regimens were not uncommon, however, platinum-based and plant alkaloid regimens were rare (7% 97/1382 and 2% 23/1382 respectively) (Appendix A). Combination therapy was also observed in 23% (312/1382). Preoperative radiotherapy was a component of NAT in 51% (868/1699) patients. A minority of patients in a single study undergoing solely radiotherapy as the NAT regime (28%, 31/112) [40] was included under NAT cohort as outcomes were reported for all patients.

Adjuvant therapy was reported in 14 studies [7,25,26,27,28,37,41,42,44,45,47,48,51,52] and was received by 66% of patients (1649/2505). This included either chemotherapy (61%, 1011/1649) or chemoradiotherapy (39%, 638/1649). In one study [52], only those with adjuvant therapy vs. neoadjuvant was compared and those with US without adjuvant therapy were excluded as outcomes were not reported.

### 3.2. Resection and Resectability in the NAT Cohort

Following NAT, 1699 proceeded to resection (87%). Failure to proceed to resection was due to cancer progression during NAT (35%, 86/243) or severe NAT toxicity leading to disruption of therapy (11%, 27/243). However, the indication for non-operative management was not reported in 53% of patients (130/243, Figure 2). The indication for resection among BRPC and LAPC was reported in nine studies. BRPC was included for resection in patients with the superior mesenteric artery (SMA) or superior mesenteric vein (SMV) involvement not exceeding 180 degrees in six studies [28,38,43,45,51,52] and without tumour extension in the celiac axis unless considered safe for complete resection in two studies [28,38]. Those without SMA involvement and < 180-degree involvement of SMV was the criteria for resection in one study [29]. LAPC was considered resectable in patients with common hepatic artery, root of splenic artery or celiac axis involvement in one study [26], and SMA or coeliac axis (CA) involvement with stenosis or thrombosis of the vessel in another study [42]. Four studies [6,7,25,28] defined BRPC by the national cancer network guidelines [55]. The types of resection were reported in 17 studies [6,8,25,26,28,29,37,40,41,42,43,45,48,49,50,51,53]: pancreaticoduodenectomy (PD, including pylorus preserving variant, 79%, 2639/3344), distal pancreatectomy (17%, 564/3344), and total pancreatectomy (4%, 141/3344).

The preoperative tumour resectability for all patients undergoing resection was reported in 17 studies [6,7,8,26,27,28,29,38,41,43,45,47,48,49,50,52,53] for NAT and US groups. BRPC were the most prevalent (47%, 1074/2264), followed by RPC (42%, 949/2264) and locally advance pancreatic cancer (LAPC) (10%, 230/2264). In the NAT group, 15% (131/866) were LAPC, 23% (203/866) RPC, and 60% (522/866) were BRPC. In the US group, 7% (99/1398) were LAPC, 39% (552/1398) BRPC, and 53% were (746/1398) RPC. Eleven patients had unresectable tumours that subsequently underwent resection in three studies [26,42,52] (NAT 10/866, US 1/1398, Appendix A).

### 3.3. Primary Outcome Measure

#### Sites of Recurrence

The weighted locoregional recurrence rate was lower amongst patients receiving NAT first, 12% and 27% (Figure 3A) in patients under NAT and US, respectively (OR 0.39 (CI 0.22–0.70); *p* = 0.004, Figure 4A) in 16 studies [7,25,28,38,40,41,42,43,45,46,49,50,51,52,53,54]. On subgroup analysis of all studies published after 2010, consistent improvement in locoregional recurrence with NAT was observed (*p* = 0.014, Figure 4A). The weighted locoregional recurrence rate in BRPC undergoing resection was also improved with NAT (NAT 25.8% (CI 20.9–31.2), Figure 3B) compared to US (US 37.7% (CI 30.4–45.4) OR 0.62 (CI 0.44–0.87), *p* = 0.007) in four studies [7,28,38,45]. The locoregional recurrence rate for RPC in the US was 11.9% (CI 7.5–17.1) in four studies [42,43,49,53]. This analysis could not be compared for RPC in the NAT group due to being only reported in two studies [49,53]. NAT was associated with a lower weighted liver recurrence rate (19.4% CI 13.4–27.2) compared to US (30.1% CI 22.7–38.7 Figure 3A) (OR 0.55 (CI 0.34–0.89), *p* = 0.023, Appendix A) in eight studies [7,25,28,40,41,43,46,49]. Weighted lung and peritoneal recurrence rates did not differ between NAT and US cohorts (*p* = 0.705 and *p* = 0.549 respectively, Figure 3A and Appendix A), and the overall rate of distant recurrence was similar between NAT and US in 13 studies [7,25,28,38,40,42,43,45,46,49,50,51,53] (NAT 46%, 463/997 US 51% 1003/1962, OR 0.92 (CI 0.61–1.39), *p* = 0.664, Figure 4B). Similarly, no difference was observed on subgroup analysis of studies published following 2010 (Figure 4B).

Similar weighted rates of distant recurrence were observed for BRPC in the NAT (NAT 50.9% CI 38.8–62.8) and US groups (US 54.7% CI 38.7–70.2), Figure 3B) in four studies [7,28,38,45]. The rate of distant recurrence for RPC in the US was 36.6% (CI 19.6–55.5) in four studies (Figure 3B). Again, this could not be compared to RPC in the NAT group as this was only reported in two studies [49,53]. The most common site of recurrence was the liver for BRPC and RPC in the US (21% 280/1342 and 53% 114/216 respectively) and NAT (9% 49/534 and 28% 65/231 respectively) groups in six [7,25,28,38,45,48] and five studies [42,43,49,50,53], respectively. The remaining distribution of resectability-specific organ sites were infrequently reported. On subgroup analysis of six studies [7,28,38,45,49,53] reporting cohorts comprised solely of BRPC or RPC, no significant differences were found for locoregional (*p* = 0.166) and distal recurrence rates (*p* = 0.786) (Appendix A).

Site-specific recurrence was reported in at least three articles for both gemcitabine [6,40,45,49,50] and fluoropyrimidine [40,45,49,51], whereby liver recurrence was the most frequent and the distribution of organ-specific weighted recurrence rates were similar between both agents (Figure 3C). Comparisons of recurrence between different chemotherapeutic agents were not reported.

### 3.4. Secondary Outcome Measures

#### Overall Recurrence

The weighted mean follow-up interval for recurrence outcomes was 40.8 months (CI 33.4–48.1). During this interval, the absolute overall recurrence rate for the entire cohort was 70% (2345/3351, Figure 2) reported in 16 studies [7,25,28,38,40,41,42,43,45,46,49,50,51,52,53,54]. The weighted overall recurrence rate was 63.4% (CI 51.8–73%) for NAT, significantly lower than the 74% (CI 68.7–80%) weighted overall recurrence rate of the US cohort (OR 0.67 (CI 0.52–0.87), *p* = 0.006, Figure 5A). On subgroup analysis of RPC, consistent improvement in weighted overall recurrence rates with NAT was observed in five studies [42,43,49,50,53] (NAT 50% 115/231, US 56% 122/216, OR 0.68 (CI 0.50–0.94), *p* = 0.029). However, for BRPC, similar weighted overall recurrence rates between NAT and US were observed on subgroup analysis of six studies [7,25,28,38,45,48] (NAT 75% 400/534 US 80% 1068/1342, OR 0.61 (CI 0.28–1.31), *p* = 0.158, Appendix A). The presence or absence of preoperative chemo radiotherapy in the NAT cohort did not impact the significant difference in weighted overall recurrence rate (with preoperative chemoradiotherapy 66% (CI 54.6–75.8) and without 68.9% (CI 53.4–81), Appendix A). On meta-regression, borderline resectability, perineural invasion and N0 nodal status were independent predictors of overall recurrence (Appendix A).

### 3.5. Time to First Recurrence

NAT was associated with a significantly longer weighted mean time to first recurrence (18.8 months, CI 14.5–23.5) compared to US (15.7months, CI 12.2–19.2) in nine studies [6,26,27,37,39,40,41,45,47] (OR 0.18 (CI 0.05–0.32), *p* = 0.015, Figure 5B). Fewer than three studies reported resectability-specific time to first recurrence in both the US and NAT groups and no study reported organ-specific time to first recurrence. The weighted time to first recurrence for each specific NAT agent ranged from 18.5–22.9 months (Figure 3D). The weighted time for each individual agent was calculated from studies using that specific NAT as the base or in combination with another agent. Two studies were excluded in the meta-analysis of time to first recurrence as outliers significantly skewed mean estimates for the US cohort [51] and the significant discrepancy in cohort numbers made the output uninterpretable [25].

### 3.6. Recurrence-Free Survival

The weighted RFS rate at 2 and 5 years was 40.5% (CI 29.8–52.2%) and 22.2% (CI 9.2–44.5%) in the NAT cohort and 24.9% (CI 17.7–33.9%) and 13.4% (CI 7.8–22.1%) in the US respectively. NAT was associated with a significantly higher RFS rate at both time points (Two year: OR 1.79 (CI 1.23–2.61), *p* = 0.005 and Five year: OR 1.95 (CI 1.03–3.69), *p* = 0.043 respectively) (Appendix A) in 15 [26,28,29,37,39,40,41,42,44,45,47,48,49,50,52] and 9 studies [8,29,40,41,45,47,48,50,52] respectively. One study [44] only reported 2-year RFS data for those who began NAT but not specific to those who ultimately underwent resection.

### 3.7. Meta-Analysis of Pathological Outcomes between NAT and Upfront Surgery

The post-resection pathology is summarised in Appendix A. NAT was associated with a smaller tumour diameter (SMD-0.67 (CI 1.05–0.28), *p* = 0.002, Appendix A). This correlated with a lower rate of perineural invasion in 12 studies [6,25,27,28,37,41,42,43,44,45,49,51] (NAT 53%, 392/744 US 80% 1504/1886, OR 0.30 (CI 0.18–0.49), *p* < 0.001, Appendix A), lymphovascular invasion in 14 studies [6,7,25,26,27,28,37,38,43,44,45,48,49,51] (NAT 43%, 394/909 US 52%, 1014/1966, OR 0.50 (CI 0.25–0.98), *p* = 0.044, Appendix A), higher rates of N0 nodal status in 21 studies [6,7,8,25,26,27,28,29,37,38,41,42,43,44,45,46,47,48,49,50,53] (NAT 58%, 840/1441 US 29% 847/2939, OR 3.36 (CI 1.93–5.84), *p* < 0.001, Appendix A) and higher rates of R0 resection in 22 studies [6,7,25,26,27,28,29,37,38,40,41,42,43,44,45,47,48,49,50,51,53,54] (NAT 78%, 1120/1434 US 67% 1909/2837, OR 1.86 (CI 1.27–2.71), *p* = 0.003, Appendix A) compared to those undergoing US. However, comparable rates of poor tumour differentiation in 11 studies [6,8,25,27,28,29,40,41,42,45,50] (*p* = 0.494, Appendix A) were observed.

### 3.8. Heterogeneity and Risk of Bias

Moderate heterogeneity was found in the outcomes to assess locoregional and distant recurrence rates. Moderate heterogeneity was further reported for liver recurrence and 2- and 5-year RFS. The pathological outcomes of tumour diameter, rates of lymphovascular involvement, and N0 nodal status also observed marked heterogeneity in the dataset. Overall, the non-randomised studies scored poorly in the MINORS criteria (Median 13 Range 11–19). Significant deficiency with regards to unbiased prospective data collection, blinded assessment of outcomes, and power calculations were seen globally (Appendix A). Many studies also failed in the domain of equivalent baseline characteristics (*n* = 12) [6,25,26,28,39,41,42,43,46,50,52,54]. Cochranes risk of bias tool also identified obvious deficits in the three randomized controlled trials (RCTS) [7,44,47] in the domains of blinding of participants and outcomes, which is an obvious consequence of the nature of the disease (Appendix A). Significant publication bias was observed on visual inspection of Forest plots for the outcomes of tumor diameter, N0 nodal status, and lymphovascular invasion (Appendix A).

## 4. Discussion

Recurrence after pancreatic resection for PDAC is frequent with 20% of patients developing recurrence within 6 months and a further 40% in the first year of resection in spite of a margin-free resection [56]. Even among those who complete adjuvant therapy, median overall survival remains less than 20 months [57,58]. Recent chemotherapeutic regimens such as FOLFORINOX and Gemcitabine plus Nab-paclitaxel have shown favourable results in improving resectability even in patients with LAPC with resection rates ranging between 0–40% [59]. However, there is paucity of level 1 evidence regarding the role of NAT on recurrence patterns for resectable and BRPC. Two recent RCTs explored the role of NAT in resectable and BRPC. Jang et al. in a recent phase 2/3 RCT comparing neoadjuvant chemo radiation with gemcitabine versus upfront resection in patients with BRPC showed comparable over all recurrence rates and recurrence patterns. On the contrary, the Dutch PREOPANC phase III randomised controlled trial comparing preoperative chemo radiotherapy versus upfront surgery for resectable and BRPC showed improved disease-free survival in the NAT group although the time to recurrence and recurrence patterns were not fully evaluated. The present review of 26 studies including 4986 patients has shown that NAT was associated with lower overall recurrence rates (63% vs. 74%), lower loco-regional recurrence rates (12 vs. 27%), and particularly in patients with BRPC. NAT was associated with a lower liver recurrence rate (19.4 vs. 30.1%) and improved recurrence-free survival at 2 years (40.5% vs. 24.9%) and 5 years (22.2% vs. 13.4%). Furthermore, on analysis of resection specimens, NAT appears to reduce rates lymphovascular (43% vs. 52%) and perineural invasion (53% vs. 80%) and increase the number of resections with N0 lymph node status (58% vs. 29%). These results suggest the superiority of neoadjuvant therapy over upfront resection in reducing loco-regional recurrences, overall recurrence, time to recurrence, and improving recurrence-free survival.

In the present review, subgroup analysis of recurrence rates in BRPC patients showed comparable overall recurrence rates for BPRC in NAT and US. This is despite improved locoregional recurrence with NAT in BRPC patients. This is in direct contrast to RPC where a significant overall recurrence benefit was observed. In theory, the effect on RPC is surprising, however, this finding is similar to the preliminary overall survival results from the Prep-02/JSAP-05 trial [60] in which NAT (gemcitabine and S-1) performed better than immediate surgery for resectable PDAC with no significant survival benefit in the BRPC group. Previous reports have highlighted significant discrepancies in the outcomes following NAT for BRPC, and many authors attribute this finding to the failure to reach a consensus with regards to a standardised definition [61,62]. The recently concluded PREOPANC trial showed superior overall survival (OS) after preoperative chemo radiotherapy for BRPC and no significant difference for resectable PDAC. Although multiple societies have endeavored to provide standardised and, more importantly, validated definitions for PDAC. Assifi et al. reported that 40% of patients diagnosed as BRPC using the AHPBA/SSO or SSAT definitions could be reclassified as RPC [63]. Indeed, the degree of vessel involvement [54] and the individual vessel involved [62] (portal vein, hepatic artery, and or superior mesenteric artery) are on their own predictors of disease-free survival and so variations in definitions will undoubtedly influence rates of recurrence. BRPC is known to have higher rates of failure to achieve R0 resection, higher rates of occult metastases and requires resection of more tissue [64]. BRPC may therefore represent a spectrum of disease rather than a single entity, which may account for the conflicting results noted in these groups of patients in the literature. The inclusion of this heterogeneity in the cohort may have contributed significantly to the lack of a survival advantage with NAT. NAT is generally tolerated much better and morbidity following NAT is considered independent of surgical morbidity [65]. Eighty-nine percent of patients successfully completed NAT and underwent surgery in the review cohort, significantly higher than previously reported, which may be contributed to by the high rates of RPC undergoing NAT. This is also far superior to the 66% completed rate for adjuvant therapy reported in the literature [66].

Although chemotherapeutic subgroup analysis was limited, time to first recurrence and the distribution of NAT-specific rates of organ-specific recurrence did not differ between agents. Gemcitabine was the most frequently utilised NAT regimen in this cohort employed in nearly 60% of patients. However, following the results of the PRODIGE-24 trial [67] that demonstrated superior disease free survival (DFS) in patients with FOLFIRINOX in comparison to Gemcitabine for adjuvant therapy, FOLFIRINOX has become common practice. Indeed, combined NAT regimens including Gemcitabine and FOLFIRINOX have shown a high response rate, and there is increasing evidence to suggest that multi-agent NAT may suppress the recurrence of distant metastases [68,69]; however, to date, no trial has been published comparing FOLFIRINOX to other options. In the current review, combination therapy was only employed in 23% of patients and 13% of patients were administered FOLFIRINOX, suggesting the outcomes of NAT in this review may be more reflective of other chemotherapeutic agents. The efficacy of FOLFIRINOX is the topic of current investigation. The interim results of ESPAC-5F (ISRCTN89500674), Phase II RCT comparing immediate surgery with neoadjuvant gemcitabine plus capecitabine or FOLFIRINOX or chemoradiotherapy in patients with borderline resectable pancreatic cancer showed a significant survival benefit with NAT. The currently ongoing PREOPANC2 trial (NTR7292, 2018-06-19), NorPACT-1 trial (NCT02919787) [70], SWOG S1505 and PANACHE01-PRODIGE48 trial (NCT02959879) [71] may provide further evidence to support NAT first strategy. Furthermore, additional radiotherapy to NAT remains uncertain [72,73]. Although some reports suggest a significantly reduced rate of lymph node metastases and local recurrence with neoadjuvant chemo-radiotherapy (NACRT) [38], the subgroup analysis from this review showed no significant advantage with NACRT. Radiotherapy also has the added risk of increased postoperative morbidity and may compromise surgical outcomes [38].

In the present review, NAT was also associated with reduced tumour diameter, lower rates of perineural invasion, and lymphovascular invasion, leading to higher rates of R0 resection. This may have contributed to the lower loco-regional recurrence rates in the NAT group. However, there were comparable rates of distant recurrence between NAT and upfront resection groups, and this may be contributed to by the lack of significant differences in lung or peritoneal recurrence rates between the two groups. Whether the improved pathological parameters would translate into long term reduction in recurrence rates and survival benefit is still a matter of debate. Even in patients achieving complete pathological response after NAT, recurrence often occurs in half the patients [74].

There remain several limitations to the methodology of this review. This is a review of largely retrospective articles with limited outcomes available for pairwise comparisons. There were obvious discrepancies in the definitions of BRPC and LAPC that would have added significant heterogeneity to the cohort. There also remains significant variability in the assessment of outcomes over time, including methods of assessment for liver metastases following the increasing availability of magnetic resonance imaging (MRI). This study also reported outcomes on solely patients who underwent resection following NAT, thereby excluding those who failed to complete NAT from the final analysis. An intention to treat analysis was abandoned due to the failure of most studies to report on the original cohort. Furthermore, there remains some selection bias in favor of NAT therapy for improving rates of liver metastases in view of the exclusion of NAT patients who ultimately failed to undergo resection. This may have been due to multiple factors including local disease progression or distant metastases. Patients were also recruited from various institutions around the world with differing perioperative care protocols. The preoperative functional status of patients in each arm were also poorly reported and were therefore underutilised to compare the cohorts at baseline. Propensity score matched data was used where reported, however, this represented less than 20% of the cohort. The study data was limited in its ability to report chemotherapy agent-specific outcomes, and FOLFIRINOX-based chemotherapy regimens were uncommon despite there being a current trend towards its use in the current day. Furthermore, the recurrence rate was reported with varying definitions from when it was calculated. Significant deficiency was also identified regarding prospective data collection, blinded assessment of outcomes, and power calculations. However, this is the first review to quantitatively report the comparative recurrence outcomes including patterns of recurrence for this complex cohort of patients.

## 5. Conclusions

This is the first quantitative analysis of recurrence patterns following NAT for PDAC. The review found NAT improved rates of overall recurrence in RPC with lower locoregional and liver recurrence. However, NAT did not appear to impact peritoneal or lung recurrence. NAT was also associated with a reduced time to recurrence and recurrence free survival at both 2 and 5 years. The review is limited by its largely retrospective dataset, and FOLFIRINOX-based chemotherapy regimens are poorly reflected in the included studies. We await the results of currently ongoing RCTs and large prospective cohorts to confirm the findings of this review and compare the relative efficacy of various NAT regimens.

## Figures and Tables

**Figure 1 jcm-09-02132-f001:**
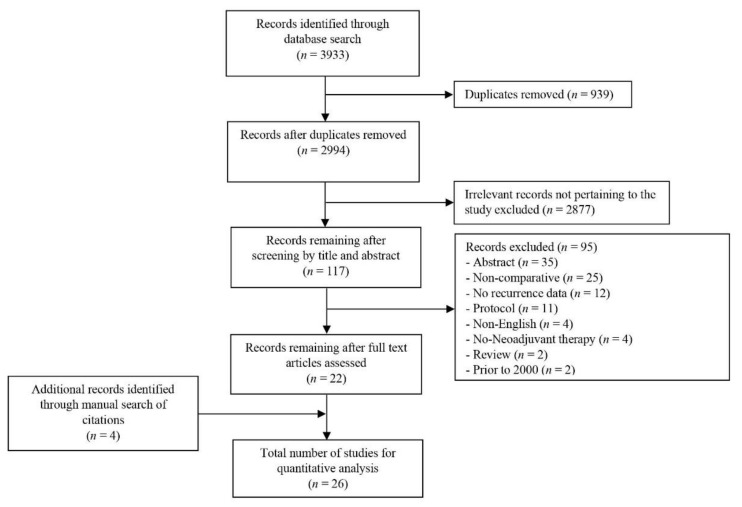
PRISMA flow chart of literature search strategy.

**Figure 2 jcm-09-02132-f002:**
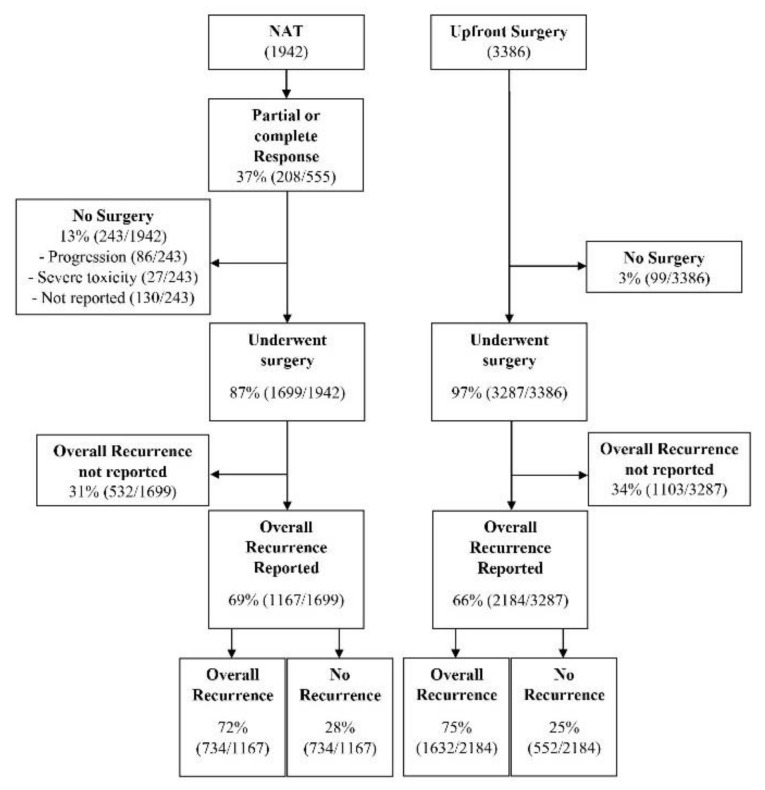
Flow chart of patient management and overall recurrence for patients initially allocated to neoadjuvant therapy (NAT) and upfront surgery.

**Figure 3 jcm-09-02132-f003:**
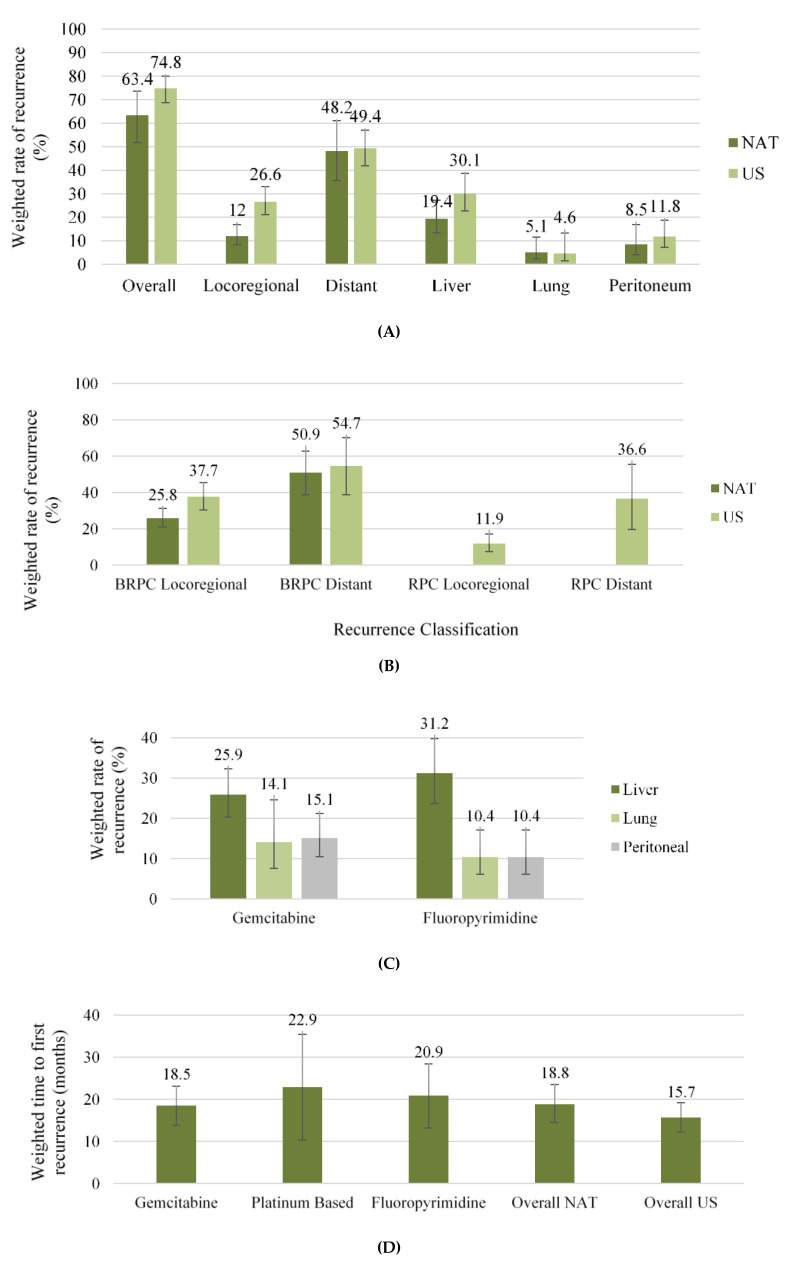
Weighted estimates of: (**A**) Site-specific rates of recurrence; (**B**) Resectability-specific sites of recurrence; (**C**) NAT-specific rates of distant organ-specific recurrence; (**D**) NAT-specific time to first recurrence. Ninety-five percent confidence intervals are denoted by the central bar. Neoadjuvant therapy (NAT); Up-front Surgery (US); Borderline resectable pancreatic cancer (BRPC); resectable pancreatic cancer (RPC).

**Figure 4 jcm-09-02132-f004:**
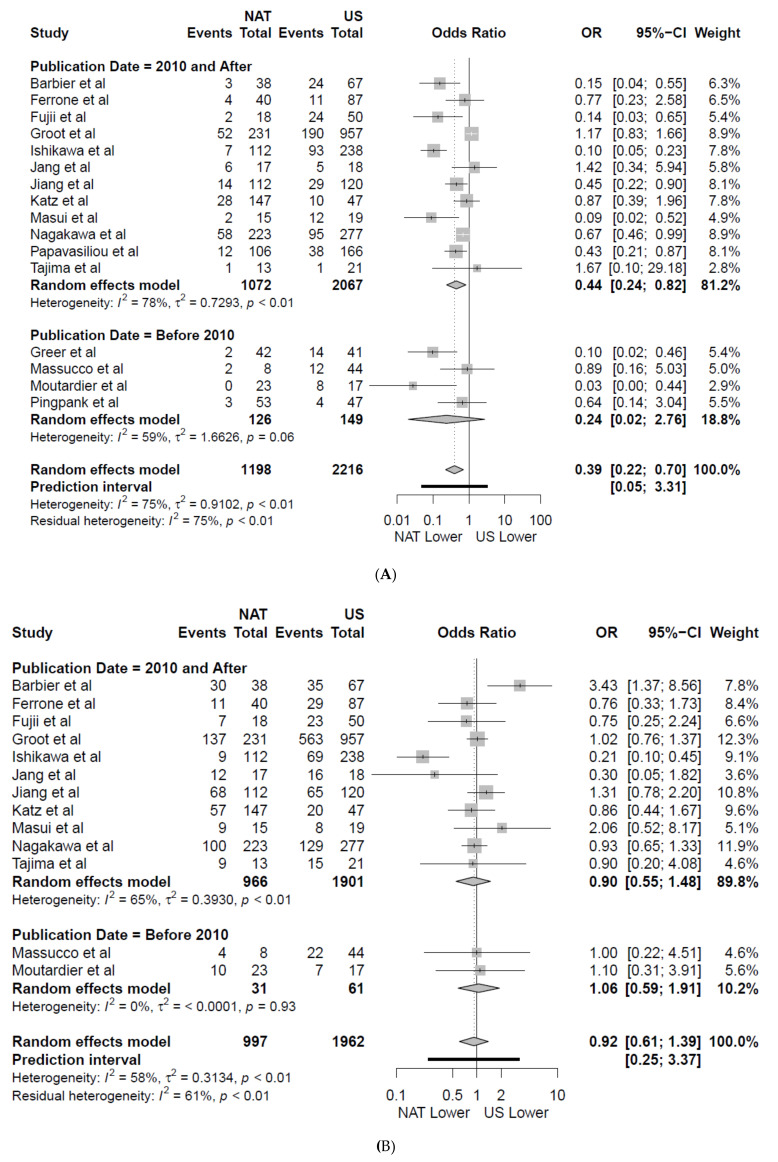
Forest plot showing: (**A**) locoregional (*n*) and (**B**) distal recurrence rates (*n*) following neoadjuvant therapy (NAT) vs. up-front Surgery (US). The results of a subgroup analysis of all articles published after 2010 are also shown. A Manel–Haenszel random effects model with a Hartung–Knapp adjustment was used for the meta-analysis of all outcomes. A Sidik–Jonkman estimator was utilised for tau^2^. Odds ratios (OR) are shown with 95 percent confidence intervals (CI).

**Figure 5 jcm-09-02132-f005:**
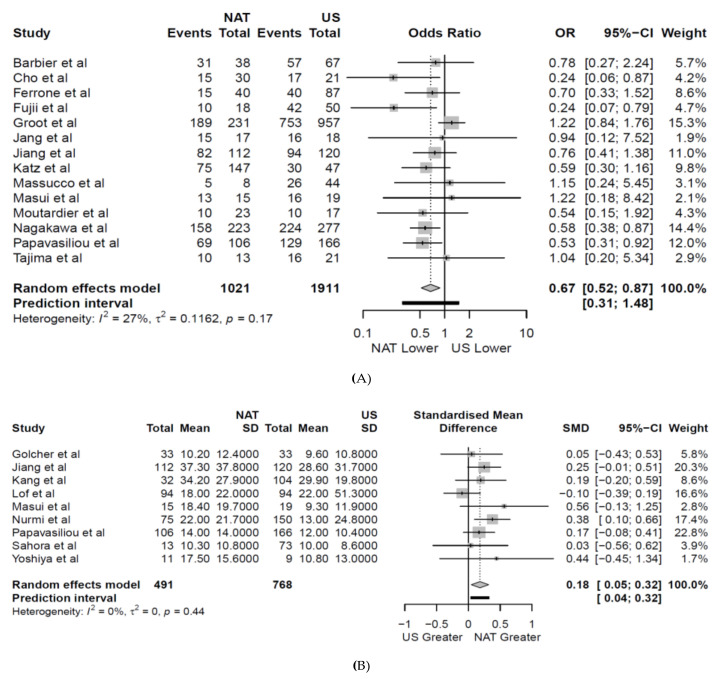
Forest plot showing: (**A**) overall recurrence rates (*n*) and (**B**) time to first recurrence (months) following neoadjuvant therapy (NAT) vs. up-front Surgery (US). A Manel-Haenszel random effects model with a Hartung–Knapp adjustment was used for the meta-analysis of all outcomes. A Sidik–Jonkman estimator was utilised for tau^2^.

**Table 1 jcm-09-02132-t001:** Characteristics of all included studies and patients recruited into the meta-analysis for quantitative analysis.

Author	Publication Year	Country	Study Type (*n*)	Recruitment Dates	Centres (*n*)	Total Study Population (*n*)	Cohorts Undergoing Surgery for Comparison		Total in Quantitative Analysis (*n*)
							NAT (*n*)	US (*n*)	
Versteijne et al. [44]	2020	Netherlands	Trial	2013–2017	16	246	72	92	164
Lof et al. [37]	2019	Europe	Retrospective	2007–2015	Multiple	1236	94 *	94 ^±^	188
Yoshiya et al. [26]	2019	Japan	Retrospective	2008–2018	1	20	11	9	20
Groot et al. [25]	2019	USA	Retrospective	2007–2015	1	1188	231	95 *	1188
Nagakawa et al. [38]	2019	Japan	Prospective	2011–2013	63	884	223	27 *	500
Nurmi et al. [27]	2018	Finland	Retrospective	2000–2015	1	225	75	150 *	225
Jang et al. [7]	2018	Korea	Trial	2012–2014	4	40	17	18	35
Chen et al. [8]	2017	China	Retrospective	2004–2013	Multiple	196	98	98 *	196
Masui et al. [45]	2016	Japan	Prospective	2005–2010	1	37	15	19	34
Ferrone et al. [43]	2015	USA	Retrospective	2011–2014	1	127	40	87	127
Fujii et al. [28]	2015	Japan	Retrospective	2002–2014	1	92	18	50	68
Ishikawa et al. [46]	2015	Japan	Retrospective	Until 2011	1	506	112	238	350
Golcher et al. [47]	2014	Germany	Trial	2003–2009	1	66	33	33	66
Papavasiliou et al. [41]	2014	USA	Retrospective	1990–2009	1	309	108	201	309
Cho et al. [48]	2013	Korea	Retrospective	2002–2011	1	51	30	21	51
Jiang et al. [40]	2013	China	Retrospective	2004–2010	1	232	11 ^β^	120	232
Kang et al. [6]	2012	Korea	Retrospective	1999–2010	1	136	32	104	136
Tajima et al. [49]	2012	Japan	Retrospective	2006–2009	1	34	13	21	34
Barugola et al. [29]	2012	Italy	Retrospective	2001–2008	1	403	41	362	403
Katz et al. [50]	2012	USA	Retrospective	2004–2008	1	147	147	47	194
Barbier et al. [51]	2011	France	Retrospective	1997–2006	1	173	38	67	105
Sahora et al. [39]	2011	Asutria	Retrospective	2003–2006	1	106	13	73 ^¥^	86
Greer et al. [52]	2008	Lebanon	Retrospective	1993–2005	1	102	42	41 ^€^	83
Massucco et al. [42]	2006	Italy	Retrospective	1999–2003	1	72	8	44	52
Moutardier et al. [53]	2004	France	Retrospective	1997–2002	1	87	23	17	40
Pingpank et al. [54]	2001	USA	Retrospective	1987–2000	1	100	53	47	100

NAT Neoadjuvant therapy; US Upfront surgery; * Propensity score matched data; ^€^ Did not include operation alone as specific outcomes were not reported for this; ^±^ Historical cohort; ^¥^ Unpublished cohort; ^β^ Included both neoadjuvant chemotherapy (*n* = 81, 72%)and neoadjuvant radiotherapy alone (*n* = 31, 28%) as outcomes were reported for overall NAT cohort.

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
