# Peer review of "Recurrence Patterns for Pancreatic Ductal Adenocarcinoma after Upfront Resection Versus Resection Following Neoadjuvant Therapy: A Comprehensive Meta-Analysis"

_jcm, 2020, doi:10.3390/jcm9072132_

Round 1
Reviewer 1 Report
This meta-analysis addresses the recurrence patterns after pancreatic resection in patients with or without neoadjuvant therapy. The authors found neoadjuvant therapy to result in lower overall recurrence rate and improved locoregional disease control.
The findings of this study are interesting and of timely relevance, however I have a few questions and remarks:
The database/studies used have several inherent biases:
The time interval spanning two decades is too long: introduction of minimally invasive techniques, newer therapies and diagnostic modalities and change in indication have an influence.
The subgroups borderline resectable carcinomas should be excluded or at least better separated in a subgroup
Liver assessment re metastasis changed massively over time (MR)
Why were studies with not reported overall recurrence included? (fig 2)
Further, I don’t think we compare the same patient group: there is a selection bias as the NAT group has 13% dropouts, probably the patients with rapid progressive disease, who are still included in the US group
A minor point, there are quite some typos, a thorough read-through will be necessary
Reviewer 2 Report
This is a meta-analysis investigating the effect of neoadjuvant therapy on the sites and patterns of recurrence after surgery for pancreatic adenocarcinoma. The authors identified 26 articles studying 4986 patients. They found that the use of neoadjuvant therapy is associated with lower overall recurrence rate especially in borderline resected PDAC and lower rate of liver metastases. This is an interesting study.
-The authors found that the receipt of neoadjuvant therapy is associated with decrease liver metastasis rate. This may be just a selection bias created by using neoadjuvant therapy and not necessary a direct causal effect from treatment. As per the recent NAT trials about 30% of patients who start neoadjuvant therapy will not undergo resection due to multiple factors including progression and development of metastasis. Since the studies included were not an intention to treat analysis, excluding these patients will favor the NAT therapy group, especially when measuring metastasis. I would suggest including this explanation in the discussion/limitation.
-Would be interesting if the authors could separate the neoadjuvant chemotherapy and chemoradiation groups if feasible.
-The authors mentioned that there is no trial that compare FOLFIRINOX to other options. Two recent trials were presented at ASCO, the ESPAC5 and SWOG S15050
Author Response
"Please see the attachment."

Round 2
Reviewer 1 Report
all issues have been adequately adressed, the restrictions remain, but are pointed out in the limitations section